# GANISP: a GAN-assisted Importance SPlitting Probability Estimator

**Malik Hassanaly[1], Andrew Glaws[1], Ryan N. King[1]**

[1] Computational Science Center, National Renewable Energy Laboratory
15013 Denver West Parkway, Golden, Colorado 80401
malik.hassanaly@nrel.gov, andrew.glaws@nrel.gov, ryan.king@nrel.gov

## Abstract

Designing manufacturing processes with high yield and strong reliability relies on effective methods for rare event estimation. Genealogical importance splitting reduces the variance of rare event probability estimators by iteratively selecting and replicating realizations that are headed towards a rare event. The replication step is difficult when applied to deterministic systems where the initial conditions of the offspring realizations need to be modified. Typically, a random perturbation is applied to the offspring to differentiate their trajectory from the parent realization. However, this random perturbation strategy may be effective for some systems while failing for others, preventing variance reduction in the probability estimate. This work seeks to address this limitation using a generative model such as a Generative Adversarial Network (GAN) to generate perturbations that are consistent with the attractor of the dynamical system. The proposed GAN-assisted Importance SPlitting method (GANISP) improves the variance reduction for the system targeted. An implementation of the method is available in a companion repository (https://github.com/NREL/GANISP).

## Introduction

Reliability analysis of design or manufacturing processes often involves the characterization of rare events since failures should be uncommon. In turn, risk analysis requires a proper estimation of the probability of rare events. Depending on the severity and the frequency of a rare event, one may decide to mitigate their effect or simply ignore it (Hassanaly and Raman 2021). For instance, defects may creep into manufacturing processes with a low probability (Escobar and Morales-Menendez 2018) that should be accurately estimated to inform planning certification and maintenance; precise frequency estimates of extreme loads are necessary to adequately design devices resilient to low cycle fatigue (Murakami and Miller 2005). If there exists a model of the system of interest that is sensitive to the distribution of conditions observed in reality, then a Monte Carlo (MC) estimator can be used to estimate probabilities. However, this can lead to unreasonable compute times for very low probability events as the MC estimator variance scales inversely with the probability being estimated (Cérou, Guyader, and

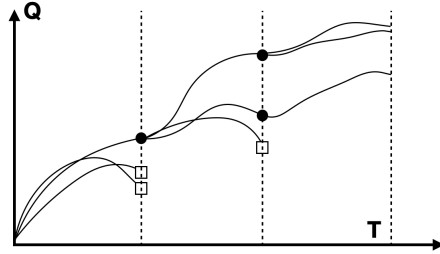

Figure 1: Graphical illustration of the genealogical importance splitting method. Selection steps are denoted by dashed lines, dots refer to cloning and squares to pruning.

Rousset 2019). This problem is exacerbated by the fact that models that approximate real systems often need to represent a wide range of scales, making each forward run expensive. It has been shown that biasing the distribution of operating conditions sampled can greatly reduce the variance of the probability estimator, which in turn reduces the number of simulations needed to estimate a rare event probability (Siegmund 1976; Glasserman et al. 1999). Importance splitting is one such approach that creates a bias towards trajectories that trend towards the desired rare event (Kahn and Harris 1951). This work focuses on a variant of importance splitting called genealogical adaptive multilevel splitting (GAMS) (Del Moral and Garnier 2005; Cérou and Guyader 2007) that can be used for deterministic systems (Wouters and Bouchet 2016; Hassanaly and Raman 2019). A graphical illustration of the method is shown in Fig. 1.

Compared to other methods like importance sampling (IS) (Siegmund 1976), it is not necessary to approximate a biasing distribution of the conditions observed by the system. In IS, poor biasing can lead to worse efficiency than MC (Cérou, Guyader, and Rousset 2019). Instead, trajectories are simulated according to the original unbiased distribution of realizations. At checkpoint locations, trajectories are then preferentially selected. The selection process of trajectories includes *pruning* non-rare trajectories and *cloning* (or resampling) rare trajectories to bias the sampled distribution towards rare events. Clones of the parent trajectory are generated to explore its neighborhood. If the system simulation is deterministic (as is the case of many modeling

approaches (Pope 2000)), then a clone that exactly copies the past parent trajectory will overlap with the parent's future trajectory and will not reduce the estimator variance. Therefore, it is necessary to apply a small perturbation to the clone's initial state (Wouters and Bouchet 2016). The primary function of the selection process is rare event probability estimation; however, this method also allows for the observation of more frequent rare events, providing greater insight into the way rare events occur (Bouchet, Rolland, and Simonnet 2019). In the context of manufacturing, observing more rare events can enable early detection of defects (Grasso and Colosimo 2017; Jenks et al. 2020).

In the rest of the paper, it is shown that in some cases the typical random cloning strategy can lead to variance reduction issues when applied to some systems. Using a generative model to perturb offspring trajectories, it is shown that this limitation can be addressed.

## Related work

### Machine learning (ML) for rare event prediction

Applications of machine learning to rare event prediction are inherently limited by the lack of data. However, encouraging results have demonstrated the ability of ML to learn useful relationships and structures from high probability data that may extrapolate to low-probability states. For example, high-probability trajectories were observed to be indicative of the low-probability path in chaotic systems (Hassanaly and Raman 2019). Additionally, the dynamics of systems learned on high probability data were shown to be useful for predicting low probability dynamics (Qi and Majda 2020), thereby enabling the use of surrogate models to accelerate the computation of rare event probability (Schöbi, Sudret, and Marelli 2017; Wan et al. 2018). In the context of importance sampling, the construction of a biasing probability density has also been facilitated by data-driven approaches (Rao et al. 2020; Sinha et al. 2020).

### Cloning strategies for importance splitting

When applied to stochastic systems, it is not necessary to perturb offspring trajectories to differentiate them from the parent. The stochastic residual of the governing equation is sufficient to prevent the parent trajectory from overlapping with its offspring. The "no-perturbation" strategy was successfully used to model zonal jet instabilities (Bouchet, Rolland, and Simonnet 2019; Simonnet, Rolland, and Bouchet 2021), drifting equation with Brownian noise (Grafke and Vanden-Eijnden 2019; Wouters and Bouchet 2016), and molecular dynamics (Teo et al. 2016). When applied to deterministic systems, random perturbations have been also been successful, such as for the Lorenz 96 equation (Wouters and Bouchet 2016; Hassanaly and Raman 2019). However, when applied to fluid flow behind a bluff body, the random perturbation strategy was observed to fail at generating diverse rare event trajectories (Lestang, Bouchet, and Lévêque 2020). A successful application of this method applied to deterministic fluid flow used perturbations applied to particular harmonics of the simulation (Ragone, Wouters, and Bouchet 2018). These combined observations suggest that random perturbation may fail for fluid flows but spatially coherent ones may be more appropriate. This motivates the present work that uses more realistic perturbations obtained with a generative adversarial network (GAN).

## Method

### Genealogical adaptive multilevel splitting (GAMS)

The proposed method builds upon the GAMS algorithm for deterministic systems (Wouters and Bouchet 2016), which is briefly described hereafter. The algorithm is suited for time-constrained systems where the quantity of interest (QoI) is defined either over a short time or at the end of a time interval $[0, T]$. The deterministic dynamical system is represented as

$$\forall\, t \in [0, T],\ \frac{d\xi}{dt} = F(\xi),\ \text{where } \xi(t = 0) \sim \mathcal{P}, \quad (1)$$

where $t$ is the time coordinate, $\xi$ is the state of the system, $F$ is the governing equation, and $\mathcal{P}$ is the distribution of the initial state for the system. Since the dynamical system is deterministic, the variability only stems from the initial condition. A quantity of interest (QoI) $Q = q(\xi)$ is chosen to define the rare event. The QoI $Q$ is a projection of the state of the system and does not entirely determine $\xi$. Given a threshold $a$ for the QoI, the probability to estimate is

$$P = Prob(q(\xi(t = T)) > a \,|\, \xi(t = 0) \sim \mathcal{P}). \quad (2)$$

To estimate $P$, one may construct an estimator $\widehat{P}$ that is unbiased, i.e., $\mathbb{E}(\widehat{P}) = P$. If the estimator is an MC estimator, its variance can be expressed as $Var(\widehat{P}) = \frac{P - P^2}{N}$, where $N$ is the number of realizations used to compute $\widehat{P}$. The relative error induced by the estimator scales as $\frac{1}{\sqrt{PN}}$. Depending on the value of the threshold $a$, the probability $P$ may be small and require a variance reduction strategy. In the GAMS method (Wouters and Bouchet 2016), multiple realizations are initially sampled from $\mathcal{P}$ and evolved over time until $t = T$. Periodically, the realizations are preferentially selected if their associated QoI is headed towards the threshold $a$. The frequency of the selection is chosen such that it is faster than the inverse of the first Lyapunov exponent, which can be efficiently calculated with two trajectories of the dynamical system (Benettin et al. 1980; Wouters and Bouchet 2016). Lyapunov exponents indicate how fast infinitesimal perturbations grow in chaotic systems, thereby overwhelming the bias introduced when cloning a realization. To determine whether a realization should be cloned or pruned, a reaction coordinate is devised and measured at every step of the simulation. As is common practice, the QoI is also the reaction coordinate (Wouters and Bouchet 2016; Lestang, Bouchet, and Lévêque 2020). The instantaneous value of the reaction coordinate along with heuristics on the most likely rare path (Hassanaly and Raman 2019) are used to dictate which realizations to clone or prune. In the original formulation of the GAMS method, a small perturbation of the form $\varepsilon\eta$ is added to every variable that defines the state of the cloned trajectories, where $\varepsilon$ is sufficiently small

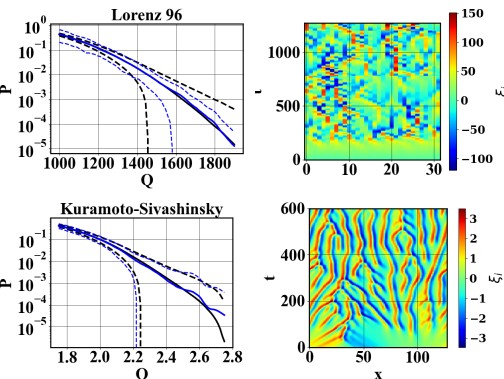

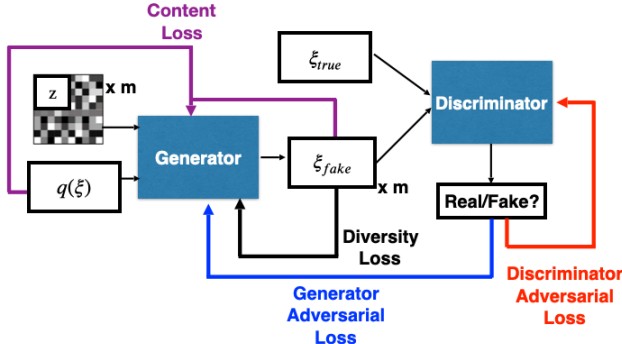

Figure 2: Application of the random cloning GAMS to the L96 equation (top) and the Kuramoto-Sivashinsky equation (bottom). Left: MC probability estimator mean (—) and standard deviation (- -) superimposed with GAMS (—) and standard deviation (- -). Right: time-evolution contour of a realization.

Figure 3: A schematic of the GANISP method, including networks and losses. In addition to the typical adversarial loss, diversity is encouraged with a diversity loss computed with a mini-batch of $m$ generated $\xi$ realizations. A content loss ensures consistency between $q(\xi)$ and $\xi$.

to not affect the probability to estimate, and $\eta$ is drawn from a standard normal distribution. This cloning technique is referred to as *random cloning*. This method is demonstrated for the 32-dimensional Lorenz 96 (L96) equation (additional numerical details are provided in Appendix) written as

$$\forall i \in [1,32], \ \frac{d\xi_i}{dt} = \xi_{i-1}(\xi_{i+1} - \xi_{i-2}) + 256 - \xi_i, \quad (3)$$

where the QoI and the reaction coordinate is

$$Q = \frac{1}{64} \sum_{i=1}^{32} \xi_i^2. \quad (4)$$

The calculations are repeated 100 times in order to quantify the variance of the probability estimator. Figure 2 shows that with random cloning, it is possible to achieve variance reduction. It is also shown in Fig. 2 that the solution of the L96 equation does not exhibit spatial coherence. In turn, it can be expected that random perturbations are consistent with the attractor of the system making random cloning well-suited for this problem.

The method is next demonstrated for the Kuramoto-Sivashinsky equation (KSE) (Kuramoto and Tsuzuki 1976; Sivashinsky 1977) (additional numerical details are provided in Appendix) written as

$$\frac{\partial \xi}{\partial t} + \nabla^4 \xi + \nabla^2 \xi + \nabla \xi^2 = 0, \quad (5)$$

where the QoI and the reaction coordinate is

$$Q = \frac{1}{128} \sum_{i=1}^{128} \xi_i^2. \quad (6)$$

In the KSE case, it is observed that the random cloning approach does not provide any variance reduction over the MC approach (see Fig. 2, bottom left). In other terms, the GAMS algorithm fails. Compared to the L96 case, the solution of the KSE exhibits stronger spatial coherence (see

Fig. 2, bottom right), which echos the failure of GAMS previously noted in a fluid flow problem (Lestang, Bouchet, and Lévêque 2020). This suggests that some systems may be better suited for random cloning and GAMS than others.

## GAN-assisted genealogical importance splitting (GANISP)

The central hypothesis in this work is that random cloning is not adequate when dealing with systems that exhibit spatial coherence. Instead, the generated clones should also exhibit spatial coherence, e.g., using a generative model. Given a parent trajectory $\xi_{parent}$ and its associated reaction coordinate $Q$, the generative model $G$ is tasked with generating solutions of the dynamical systems that have the same reaction coordinate value. This can be achieved by using a conditional Generative Adversarial Network (cGAN) (Goodfellow et al. 2014; Mirza and Osindero 2014) where the conditional variable is the reaction coordinate (see Fig. 3). This method is called GANISP.

The data used to train the model can be collected from unbiased trajectories simulated. In the GAMS algorithm, it is common to first perform a rough MC estimate to determine how to appropriately choose the number of clones to generate (Wouters and Bouchet 2016; Hassanaly and Raman 2019). These realizations are also leveraged here to collect the data used by the cGAN. In the case where the final time $T$ is sufficiently large to enter a statistically stationary state (as is the case for the KSE), each trajectory can provide multiple snapshots to train the cGAN. Additional details about the dataset are provided in the appendix. Since the GAN is trained for the statistically stationary portion of the problem (for KSE, $t > 50$), outside of that regime, it is necessary to revert to a random cloning approach.

While GANs have been shown to generate high-quality samples, they are notoriously subject to instabilities. Here, the main concern is mode collapse where the generated distribution of samples does not reflect the true support of the distribution (Salimans et al. 2016). This would hinder the ability of the cloned trajectories to sufficiently explore the

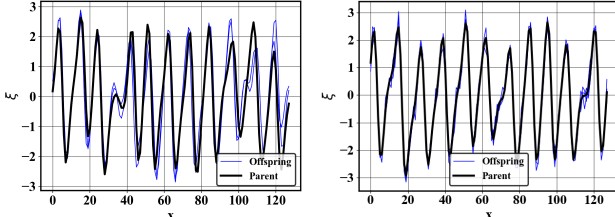
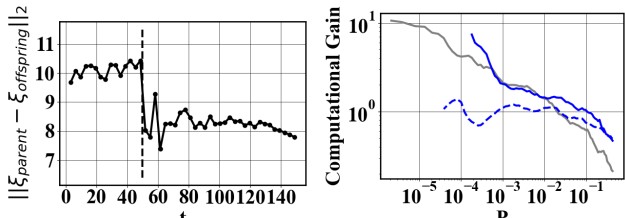

Figure 4: Example of generated samples during the cloning process (—) in comparison with the parent realization (—) for the Kuramoto-Sivashinsky equation. Left: GANISP method. Right: random cloning.

Figure 5: Left: $L_2$ norm between parent realization and clones at every selection step averaged over the clones and realizations of the importance splitting. Dashed lines denote the transition to the statistically stationary regime where the transition from random cloning to GAN-assisted cloning is operated. Right: computational gain with the random cloning technique against probability for L96 (—) and the KSE (- -), and the GANISP method applied to the KSE (—).

neighborhood of the parent simulation. Mode collapse is tackled using the method of Hassanaly et al. (2022) where one first approximates the conditional moments of the distribution $(\xi|q(\xi) = Q)$ and uses them to encourage the generation of a sufficiently diverse pool of samples. Figure 4 shows examples of generated samples along with an assessment of the diversity achieved. Additional details about the training procedure and the networks architecture are available in the appendix.

The cloning process inherently modifies the dynamics of the dynamical system which, in turn, may perturb the tail of the PDF to estimate. To mitigate this effect, the clones need to be sufficiently close to the parent realization (Wouters and Bouchet 2016). In the present case, at every cloning step, the optimization problem

$$\underset{z}{\operatorname{argmin}}||G(Q, z) - \xi_{parent}||_2 \qquad (7)$$

is solved to find the latent variable $z$ that matches the parent realization to clone $\xi_{parent}$. For computational efficiency, this problem is solved using particle swarm optimization (Karaboga and Akay 2009), which leverages the ability of the cGAN to efficiently generate batches of samples. Although the optimization increases the cost of GANISP, the added cost is marginal compared to forward runs of more expensive calculations. If $n$ clones are needed, the $n$ closest samples obtained at the end of the optimization procedure are selected. The hyperparameters of the swarm optimization are chosen such that the clones are sufficiently close to the parent realization as will be shown in the Result section. To demonstrate the importance of the optimization step, a numerical experiment is conducted in the appendix, where the optimization procedure is disabled.

## Results

Here, the benefit of GANISP is demonstrated for the KSE case, which failed when using random cloning. Before the statistically stationary part of the dynamics ($t < 50$), random cloning is used with the same magnitude as in the Method section. For $t > 50$, the cGAN is used to clone the realizations. Since the parameters of the optimization procedure from Eq. 7 dictate the magnitude of the differences between the parent and clones, the distances between the parent and offspring should be recorded to ensure that the optimization

sufficiently converged. Figure 5 (left) shows that the difference between the offspring and parent simulations was smaller when the GAN is active ($t > 50$) than when random cloning is used ($t < 50$). This demonstrates that the implementation of the optimization procedure achieves the intended goal of maintaining a small distance between parent and offspring realization.

The computational gain obtained with GAMS is computed using the ratio of the estimator variance against the MC variance for cases where the probability bias is small. Figure 5 (right) shows that, unlike the L96 case, random cloning failed at reducing the probability estimator variance for KSE. With GANISP, the estimator variance was effectively reduced and the variance reduction is similar to that obtained with the L96 problem suggesting that GANISP addressed the main limitation that affected GAMS in the KSE case. This result demonstrates that: 1) the cloning strategy does affect the performance of GAMS and 2) the generative model can effectively replace the random cloning strategy of GAMS. A notable difference that was not solved by the proposed approach is that for very small probabilities, GANISP induced as much bias as the random cloning method.

## Conclusion

In this work, a GAN-based cloning strategy is proposed to address the deficiencies of random cloning which may not be appropriate for some all systems. The proposed cloning strategy helps reduce the probability estimation variance for rare events and paves the way for the use of generative models for rare-event probability prediction. The proposed method was shown suited for the Kuramoto-Sivashinsky equation, and a more in-depth study will be needed to understand what type of system may best benefit from GANISP. Cloning inevitably disturbs the PDF to estimate and it is necessary to tightly control the magnitude of the disturbance introduced. In the present work, an optimization problem is solved to this effect and it was shown that relying on the optimization inaccuracies was sufficient and computationally efficient. More systematic and efficient optimization strategies will be devised in the future.

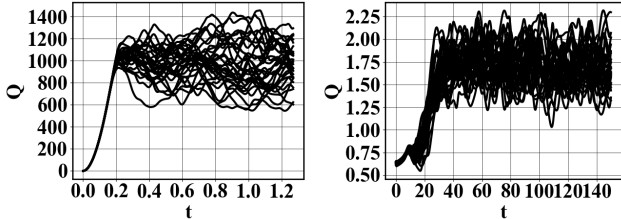

Figure 6: Time evolution of $Q$ of 30 MC realizations for Lorenz 96 (left) and the Kuramoto-Sivashinsky equation (right).

## Numerical details of the importance splitting for Lorenz 96 (L96) and Kuramoto-Sivashinsky equation (KSE)

The numerical integration of the L96 equation is done with a second-order Runge-Kutta integrator with a timestep of $dt = 0.001$ and a final time $T = 1.27$. In the KSE case, a fourth-order exponential Runge-Kutta integrator (Kassam and Trefethen 2005) is used with a timestep $dt = 0.25$ and final time $T = 150$. For the KSE, the domain is discretized in Fourier space using 128 modes that span the spatial domain $[0, 32\pi]$. The implementation of both integrators is available in the companion repository (https://github.com/NREL/GANISP).

The mean initial condition of the L96 is uniformly equal to zero and superimposed with normally distributed perturbations sampled from $\mathcal{N}(0, 1)$. For the KSE, the mean initial condition is $\cos(x/16)(1 + \sin(x/16))$ superimposed with normally distributed perturbations sampled from $\mathcal{N}(0, 0.1)$. Figure 6 shows the time evolution of $Q$ for 30 MC realizations of L96 and KSE

For the GAMS applications, the interacting particle version of the method was used (Wouters and Bouchet 2016) so that the total number of realizations simulated is held constant. For both L96 and KSE, the GAMS algorithm is run with 100 concurrent simulations. The weights assigned to each simulation (that are used to decide how many simulations are cloned or pruned) are obtained using the method of Hassanaly and Raman (2019) where the most likely average path is computed with 100 simulations. In both simulations, the target level of $Q$ is the one that corresponds to a probability of the order of $10^{-1}$ ($Q = 2.0$ for KSE and $Q = 1300$ for L96).

The cloning process is done 64 times during the L96 simulations and 45 times during the KSE equation. These frequencies were decided based on the value of the first Lyapunov exponent of the system, in agreement with the method proposed in Wouters and Bouchet (2016). For the random cloning cases, the magnitude of the noise was $\varepsilon = 0.871$ for L96 and $\varepsilon = 0.1$ for KSE. The noise magnitude was decided based so that it is the highest possible without biasing the probability estimate. The noise magnitude needs to be sufficiently large to observe rare realizations and sufficiently small to not bias the probability estimator.

## Networks architecture

The cGAN network is used as a super-resolution tool that augments the dimension of a sample from the 1-dimensional QoI value to the 128-dimensional realization $\xi$. The architecture is based on the approach of Hassanaly et al. (2022) that was originally used for multi-field super-resolution of wind data. The generator network $G(\cdot)$ receives a 16-dimensional latent variable $z$ (drawn uniformly from the interval $[-1, 1]$) and the desired 1-dimensional value of the QoI. The QoI value is augmented with a dense layer to another 16-dimensional channel. The rest of the generator network is fully convolutional and uses convolutional kernels of size 3 with parametric ReLU activations (He et al. 2015). Sixteen residual blocks with skip connections prepare the generated realizations. Super-resolution blocks increase the spatial resolution data using depth-to-space steps. The discriminator network $D(\cdot)$ is comprised of eight convolutional layers with parametric ReLU activations and two fully connected layers. The convolutional kernels of the discriminator alternate between strides size 1 and 2.

Using the method outlined of Stengel et al. (2020), a balance is maintained between the performances of the generator and the discriminator. At every step, the generator or discriminator may be trained more or fewer times if one network outperforms the other.

The dataset uses the statistically stationary part of the KSE realizations for $t > 50$ (Fig 6 right). For KSE, the integral time scale was evaluated to be $l_T = 12$ allowing to select 10 snapshots per realization. In total, 10,000 snapshots are collected from 1000 independent runs. 100 snapshots are reserved for testing and evaluating that adversarial, content, and diversity losses are correctly minimized (Fig. 7). For the proof-of-concept purpose of the paper, using this large amount of data is justified. In the future, it will be interesting to reduce the data requirement of the generative model. The training was done for 78 epochs which took 12h on a single graphical processing unit (GPU).

The generator network loss function contains three terms: (i) a content loss, (ii) an adversarial loss, and (iii) a diversity loss (Hassanaly et al. 2022). To ensure proper balancing between the losses, each term needs to be appropriately scaled. The content loss is scaled by a factor 1000, the adversarial loss by a factor 0.1, and the diversity loss by a factor 1. With these settings, the cGAN is able to generate high-quality samples (Fig. 4) while generating the appropriate diversity and consistency with the QoI (Fig. 7).

For the estimation of the conditional moments used in the diversity loss, the neural-network-assisted estimation of Hassanaly et al. (2022) is implemented. The architecture of the network used follows the generator architecture of Ledig et al. (Ledig et al. 2017) with a fully convolutional network with skip connections. Two residual blocks and fours filters are used. The neural networks (training and evaluation) were implemented with the Tensorflow 2.0 library (Abadi et al. 2016).

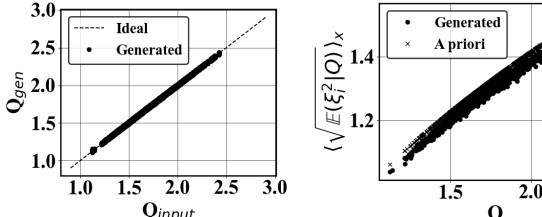

Figure 7: Demonstration of the enforcement of the generator losses. Left: enforcement of content loss. Consistency between the input QoI ($Q_{input}$) and the QoI of the generated samples ($Q_{gen}$). Right: enforcement of the diversity loss. Consistency between the a priori estimated second conditional moment averaged over space and the second order conditional moment of the generated data.

## Results with arbitrarily large perturbations

As explained in Wouters and Bouchet (2016), if the cloning process induces too large perturbations, it may bias the probability estimator. The cloned realizations are chosen sufficiently close to the parent realization to avoid this effect. In the GANISP method, the same concerns have motivated solving an optimization problem to generate clones sufficiently close to the parent realization (Eq. 7). To clearly show the importance of the optimization process, the probability estimated with GANISP for the KSE case is shown in Fig. 8 when the optimization is not used for selecting clones close to the parent realization. In that case, it can be seen that the probability estimate is biased and that the distance between the parent and cloned realizations becomes large when the GAN-assisted cloning is operated ($t > 50$).

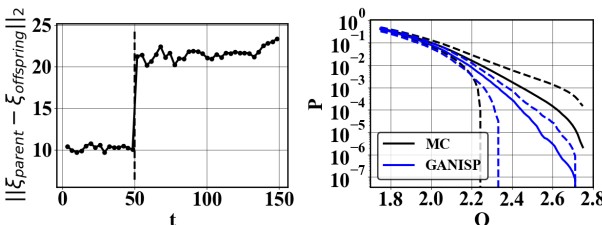

Figure 8: Left: $L_2$ norm between the parent realization and the clones at every selection step averaged over the clones and realizations of GANISP without the optimization. Dashed line denotes the transition to statistically stationary time where transition from random cloning to GAN-assisted cloning is operated. Right: probability computational gain with the random cloning technique against probability for (—) and the KSE (—), and the GANISP method applied to the KSE (- -). Right: MC probability estimator mean (—) and standard deviation (- -) superimposed with the GANISP estimator without optimization (—) and standard deviation (- -).

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

## Acknowledgments

This work was authored by the National Renewable Energy Laboratory (NREL), operated by Alliance for Sustainable Energy, LLC, for the U.S. Department of Energy (DOE) under Contract No. DE-AC36-08GO28308. This work was supported by funding from DOE's Advanced Scientific Computing Research (ASCR) program. The research was performed using computational resources sponsored by the Department of Energy's Office of Energy Efficiency and Renewable Energy and located at the National Renewable Energy Laboratory. The views expressed in the article do not necessarily represent the views of the DOE or the U.S. Government. The U.S. Government retains and the publisher, by accepting the article for publication, acknowledges that the U.S. Government retains a nonexclusive, paid-up, irrevocable, worldwide license to publish or reproduce the published form of this work, or allow others to do so, for U.S. Government purposes.
