# OpenReview forum: "GANISP: a GAN-assisted Importance SPlitting Probability Estimator"
_AAAI.org/2022/Workshop/ADAM — AAAI 2022 Workshop ADAM_

### Official Review · Reviewer_tN7B · 2021-11-29
**A GAN based improved on rare event estimation methods.**

**Rating:** 7
**Confidence:** 4

**Review:**

This paper primarily concerns methods for reducing the variance in importance splitting-based probability estimators in the context of rare events. The paper argues that the standard strategy in Genealogical importance splitting—random splitting—can fail in cases where the probability distribution has the type of degeneracies seen in rare event distributions. To get around this, the paper proposes generating perturbations instead via a GAN. Specifically, the method builds upon Genealogical adaptive multilevel splitting (GAMS) which samples small perturbations from a normal distribution ($\eta$). This paper instead substitutes $\eta$ with conditional GAN-generated samples.

Overall, this paper fits the scope of the workshop and presents some interesting ideas that could benefit discussion at the workshop. The background and motivation were well explained and the experiments were sound and easily reproducible using known examples. My only concerns, though these are minor for a workshop paper and are just suggestions for if the authors take this work further forward are: (1) It would have been nice to see this applied to a more design or manufacturing-oriented example, rather than the KSE example given, and (2) the paper only compared the KSE and KSE+GANISP method—are there other competing approaches that this method would be benchmarked well against?
One minor technical notes of possible future interest to the authors: The optimization is Equation 7 uses PSO to match samples in the latent space coordinates. You could consider in future work either (a) backpropagating directly through the generator to minimize Eqn. 7 or (b) investigating bi-directional maps between $z$ and $\xi_{parent}$ such as normalizing flows or autoencoder type models.

---

### Official Review · Reviewer_jW57 · 2021-12-01
**The authors have presented a GAN-based cloning strategy that helps in reducing the probability estimation variance for rare events. An optimization problem is solved to control the disturbance induced by cloning on the density function.**

**Rating:** 9
**Confidence:** 3

**Review:**

The advantages of GAN-based cloning strategies over random cloning have been clearly demonstrated, which is a strong point of novelty and significance of this work.

Some comments/questions regarding the approach:
1) Could you elaborate on how the hyperparameter of the swarm optimization are chosen that ensures the proximity of the clones to the parent realizations?
2) The authors have noted the large data requirement for the generative model. In case the generative model is not well trained under data limitations, what effect does that have on the optimization step? Is there a way to update the generative model sequentially during the optimization for an iterative improvement?

Overall, the work is of high quality and has great potential for growth and significance in the field.